# Innervation modulates the functional connectivity between pancreatic endocrine cells

**Yu Hsuan Carol Yang[1,2]\*, Linford JB Briant[3], Christopher A Raab[1], Sri Teja Mullapudi[1], Hans-Martin Maischein[1], Koichi Kawakami[4], Didier YR Stainier[1]\***

[1]Department of Developmental Genetics, Max Planck Institute for Heart and Lung Research, Bad Nauheim, Germany; [2]Exeter Centre of Excellence for Diabetes Research, Institute of Biomedical and Clinical Science, University of Exeter Medical School, Exeter, United Kingdom; [3]Oxford Centre for Diabetes, Endocrinology and Metabolism, Radcliffe Department of Medicine, University of Oxford, Oxford, United Kingdom; [4]Division of Molecular and Developmental Biology, National Institute of Genetics, Department of Genetics, SOKENDAI, Mishima, Japan

**Abstract** The importance of pancreatic endocrine cell activity modulation by autonomic innervation has been debated. To investigate this question, we established an in vivo imaging model that also allows chronic and acute neuromodulation with genetic and optogenetic tools. Using the GCaMP6s biosensor together with endocrine cell fluorescent reporters, we imaged calcium dynamics simultaneously in multiple pancreatic islet cell types in live animals in control states and upon changes in innervation. We find that by 4 days post fertilization in zebrafish, a stage when islet architecture is reminiscent of that in adult rodents, prominent activity coupling between beta cells is present in basal glucose conditions. Furthermore, we show that both chronic and acute loss of nerve activity result in diminished beta–beta and alpha–beta activity coupling. Pancreatic nerves are in contact with all islet cell types, but predominantly with beta and delta cells. Surprisingly, a subset of delta cells with detectable peri-islet neural activity coupling had significantly higher homotypic coupling with other delta cells suggesting that some delta cells receive innervation that coordinates their output. Overall, these data show that innervation plays a vital role in the maintenance of homotypic and heterotypic cellular connectivity in pancreatic islets, a process critical for islet function.

**\*For correspondence:**
Y.C.Yang@exeter.ac.uk (YHCY);
Didier.Stainier@mpi-bn.mpg.de
(DYRS)

## Editor's evaluation

The role of islet innervation on endocrine cell activity is currently not well defined. Previously, Yang et al. described embryonic islet innervation dynamics in zebrafish and the role of neural activity in proper glucose homeostasis and δ-cell formation. In their new manuscript, the authors introduced a novel transgenic model for whole islet in vivo calcium imaging. By applying this tool in a series of innovative and technically challenging approaches, the authors provide novel insights into how neuronal inputs influence pancreatic endocrine cell connectivity and function. Overall, this is an important study that adds to our understanding for the role of neuronal interactions with the islet.

## Introduction

Tight regulation of hormone release from pancreatic islets is critical for glucose homeostasis and its disruption can lead to diabetes mellitus (*Noguchi and Huising, 2019*). Pancreatic islets are composed

of different cell types, including the hormone producing alpha, beta, and delta cells, peripheral nerves, and vascular endothelial and smooth muscle cells. Studies have implicated signaling from the vascular scaffold (*Almaça et al., 2014*; *Mullapudi et al., 2019*) and nerve networks (*Rodriguez-Diaz et al., 2012*; *Taborsky et al., 1998*; *Borden et al., 2013*; *Yang et al., 2018*; *Makhmutova et al., 2019*; *Tarussio et al., 2014*) during the development and function of pancreatic islet cells. However, it remains difficult to investigate the immediate effects of acute nerve modulation on islet cell function. Given the alterations in islet innervation architecture in some models of diabetes (*Mundinger and Taborsky, 2016*; *Mundinger et al., 2016*; *Tang et al., 2018*), it is imperative to understand whether disruption of nervous control can contribute to diabetes etiology.

Different methods of assessing islet cell function have provided important clues into the role of autocrine and paracrine signaling in this process. Electrophysiological recordings have provided fundamental insights into isolated islet cell function (*Vergari et al., 2020*; *Camunas-Soler et al., 2020*; *Hastoy et al., 2018*), including functional connectivity studies that identified homotypic as well as heterotypic coupling between endocrine cells (*Moreno et al., 2005*; *Briant et al., 2018*). However, assessing islet function in live animals with undisrupted vascular and nerve networks remains challenging. Calcium dynamics is a good readout of the function of all islet cell types because its influx is critical for hormone release. However, no studies to date have been able to record simultaneously the activity of all islet endocrine cell types in the intact organ of a living animal, which is required to understand how the different endocrine cell types respond to physiological perturbations individually and interdependently. To this end, we established an in vivo imaging platform to visualize the activity of all the islet cell types by combining calcium imaging with cell type reporters. We investigated the functional connectivity between homotypic and heterotypic cell pairs by analyzing the correlation patterns in their intracellular calcium changes. Chronic and acute inhibition of nerve activity captured its dynamic control of the functional connectivity between islet endocrine cells.

## Results and discussion

### The activity of all pancreatic endocrine cell types can be studied simultaneously in vivo

The zebrafish primary islet becomes highly innervated (*Yang et al., 2018*) and vascularized (*Mullapudi et al., 2019*; *Hen et al., 2015*; *Toselli et al., 2019*) early in development (*Figure 1A*). Fluorescent reporters for different pancreatic endocrine cell types, including beta, alpha, and delta cells, were used to study the establishment of islet cytoarchitecture (*Figure 1B*). By 100 hours post fertilization (hpf), a beta cell core and alpha cell mantle layout are observed (*Figure 1B, C*), in agreement with previous studies (*Biemar et al., 2001*), and reminiscent of adult rodent islets (*Brereton et al., 2015*) and small human islets (*Bonner-Weir et al., 2015*). Simultaneous functional assessment of all islet cell types in vivo required reporters for cell activity and cell identity. We used the *Et(1121A:GAL4FF)* enhancer trap line with the *Tg(UAS:GCaMP6s)* line for calcium imaging of all islet cells and a subset of peri-islet neurons (*Figure 1D, E*, *Figure 1—video 1*), as well as the *Tg(ins:mCardinal)* and *Tg(sst2:RFP)* lines to assign beta and delta cell identities, respectively; alpha cells were identified by their mantle localization and/or by immunostaining (*Figure 1E*). Thus, for the first time, we were able to assess the activity of all islet cells along with some peri-islet neurons in their native environment in an intact living animal (*Figure 1F*, *Figure 1—video 2*).

### A subset of islet cells display activity coupling with peri-islet neurons

Pancreatic islet innervation is in contact with all islet cell types (*Figure 2A, B*). However, both the beta and delta cells have a higher density of nerves in contact with their cell surfaces (*Figure 2C*). Prior to 120 hpf, this innervation is only from the vagus nerve (*Yang et al., 2018*). To investigate whether peri-islet neurons could actively modulate intra-islet coordination of activity, we first used the *Et(1121A:GAL4FF); Tg(UAS:GCaMP6s)* line which also labels a subset of peri-islet neurons (*Figure 1—video 1*) to simultaneously image the calcium dynamics in neurons and islet cells. While a higher number of neural activity connection to beta cells were observed, normalization to the number of cells imaged revealed no significant differences in the percentage of beta, delta, and alpha cells that display activity coupling with peri-islet neurons (*Figure 2D, E*). From the normalized single-cell calcium traces, correlation matrices, and average correlation coefficients ($R_{avg}$), we observed that

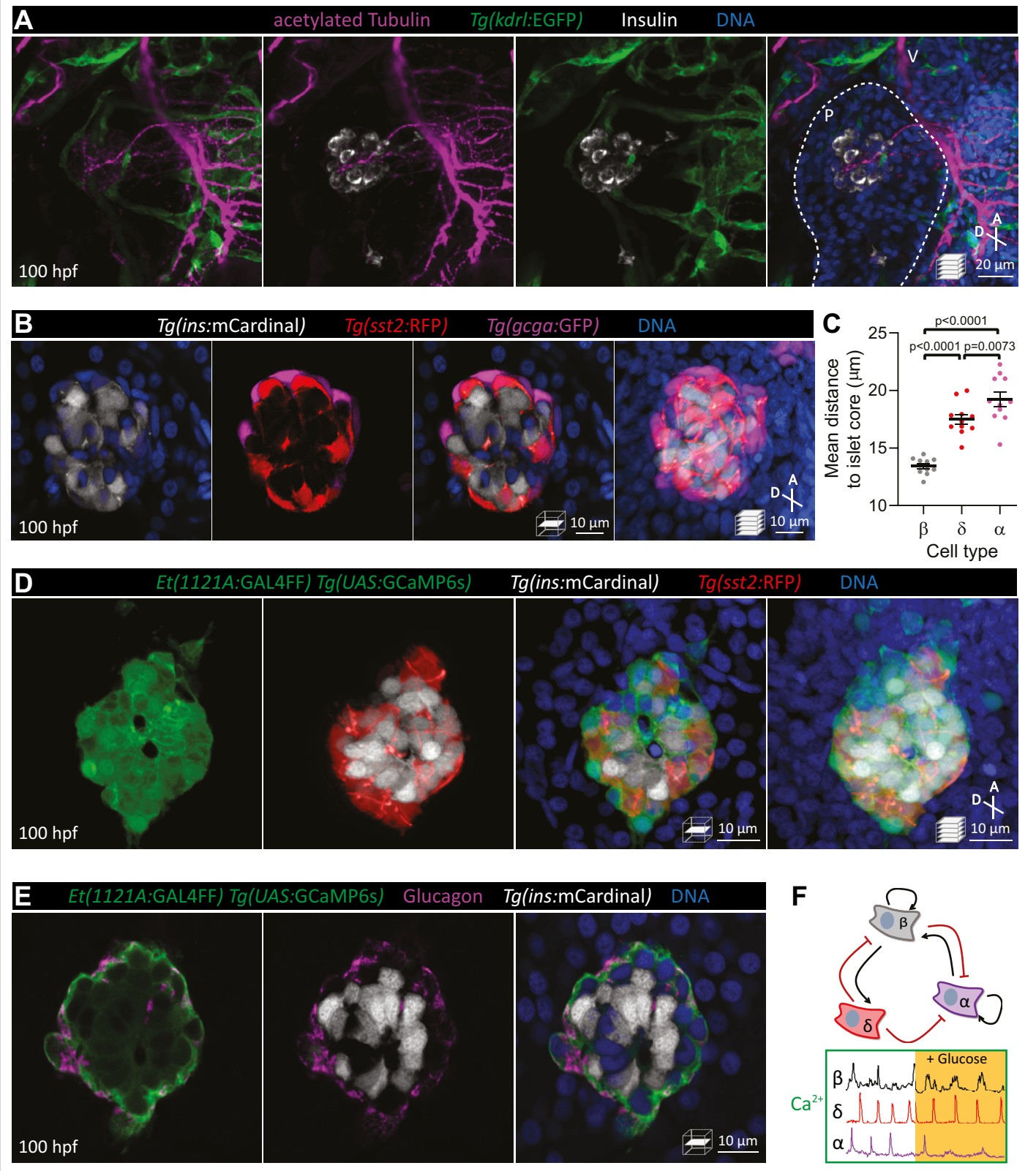

**Figure 1.** Pancreatic islet cell activity is visualized in vivo with preserved vascular and neural networks.
(**A**) Wholemount immunostaining of wild-type zebrafish at 100 hours post fertilization (hpf) for acetylated Tubulin (nerves), *Tg*(*kdrl*:GFP) expression (vessels), and Insulin (beta cells), and counterstaining with DAPI (diamidino-2-phenylindole, DNA). (**B**) 100 hpf *Tg(ins:mCardinal); Tg(sst2:RFP); Tg(gcga:GFP)* zebrafish stained with DAPI (DNA). (**C**) Mean distance of pancreatic islet cells to islet core reveals a beta cell core and alpha cell

*Figure 1 continued on next page*

Figure 1 continued

mantle; mean ± SEM, *n* = 11 animals, p values from one-way analysis of variance (ANOVA) with Holm–Sidak's multiple comparisons test; see *Figure 1—source data 1*. (**D**) 100 hpf *Et(1121A:GAL4FF); Tg(UAS:GCaMP6s); Tg(ins:mCardinal); Tg(sst2:RFP)* zebrafish stained with DAPI (DNA). (**E**) 100 hpf *Et(1121A:GAL4FF); Tg(UAS:GCaMP6s); Tg(ins:mCardinal)* zebrafish stained for Glucagon (alpha cells) and DNA. (**F**) Schematic of documented interactions between beta, delta, and alpha cells and of intracellular calcium recordings in each of these cell types. Maximum intensity projections or single planes are presented; A, anterior; D, dorsal; V, vagus nerve; P, pancreas.

The online version of this article includes the following video and source data for figure 1:

**Source data 1.** *Figure 1C*.

**Figure 1—video 1.** 100 hpf *Et(1121A:GAL4FF); Tg(UAS:GCaMP6s); Tg(ins:mCardinal); Tg(sst2:RFP)* zebrafish stained with DAPI (DNA).
https://elifesciences.org/articles/64526/figures#fig1video1

**Figure 1—video 2.** Intracellular calcium dynamics in pancreatic islet cells (including delta, beta, alpha, and unidentified cells) in 100 hpf *Et(1121A:GAL4FF); Tg(UAS:GCaMP6s); Tg(ins:mCardinal); Tg(sst2:RFP)* zebrafish.
https://elifesciences.org/articles/64526/figures#fig1video2

homotypic coupling between beta cells is more prominent than those between delta and alpha cells (*Figure 2F–H*). Notably, we found a significant increase in homotypic coupling for the delta cells that display neural activity connection (*Figure 2H*). Future studies will determine whether direct neural activity connection is critical for the regulation of this delta cell subset.

## Homotypic and heterotypic coupling between endocrine cells requires pancreatic innervation

Activity coupling between pancreatic endocrine cells can be mediated by autocrine and paracrine signaling, gap junctions, and other means. To investigate whether pancreatic innervation is critical for intra-islet coordination of activity, we used different approaches to chronically or acutely inhibit neural signaling. We used endoderm transplantation to generate chimeric zebrafish that express two GAL4/UAS systems in different germ layer-derived tissues and investigated the role of chronic neural inhibition on islet function (*Figure 3A*). Pan-neural expression of botulinum toxin (BoTx) chronically inhibits neurotransmitter release (*Yang et al., 2018*; *Sternberg et al., 2016*) and leads to elevated glucose levels at 100 hpf (*Yang et al., 2018*; *Figure 3B*). While primary islet volume was consistently greater in BoTx$^+$ larvae at 100 hpf (*Figure 3C*; as we reported for earlier stages *Yang et al., 2018*), we did not observe changes in the architectural arrangement of the different islet cell types (*Figure 3D*). From the normalized single-cell calcium traces, correlation matrices, and average correlation coefficients ($R_{avg}$), we observed that the calcium dynamics in BoTx$^+$ larvae were significantly disrupted, with impairment in beta cell coupling under both basal glucose and glucose stimulated conditions (*Figure 3E–G*, *Figure 3—figure supplement 1*). Our measure of $R_{avg}$ over increasing intercellular distance revealed the expected decline in coupling over distance, and the significant difference in the elevation of the linear regression further confirmed the altered synchronicity between beta cells (*Figure 3H*).

To determine how perturbations in neural signaling influenced communication between different endocrine cell types, we conducted correlation analysis over increasing distance as well as fraction time analysis of heterotypic cell pairs that were in nearest proximity to each other (*Figure 3H–K*, *Figure 3—figure supplement 2*). Significant changes in the intercepts suggest impairments in delta–beta and alpha–beta heterotypic coupling (*Figure 3H*). Nearest neighbors have a greater likelihood of displaying coupling, as seen in our $R_{avg}$ over increasing distance analysis. We analyzed single-cell calcium traces and determined the fraction of time a given nearest cell pair resides in a state when (1) both cells are active, (2) both cells are silent, and (3) one cell is silent, the other active. This analysis of activity patterns between nearest heterotypic cell pairs further supported the observed heterotypic coupling defects upon chronic neural inhibition (*Figure 3I–K*). Aside from the quiet phase, when both cell types are silent, we found changes in activity patterns between alpha–beta, delta–beta, and alpha–delta cell pairs (*Figure 3I–K*): upon chronic neural inhibition, the delta-silent/beta-active state was decreased while the delta-active/beta-silent state and the delta-active/alpha-silent state were both increased (*Figure 3K*). Although we cannot exclude potential defects in endocrine cell development, as reported in our previous study (*Yang et al., 2018*), these changes were not accompanied by alterations in delta cell calcium oscillation frequency nor peak height or duration (*Figure 3—figure supplement 3*). We also found a significant decrease in the alpha-active/beta-active state upon

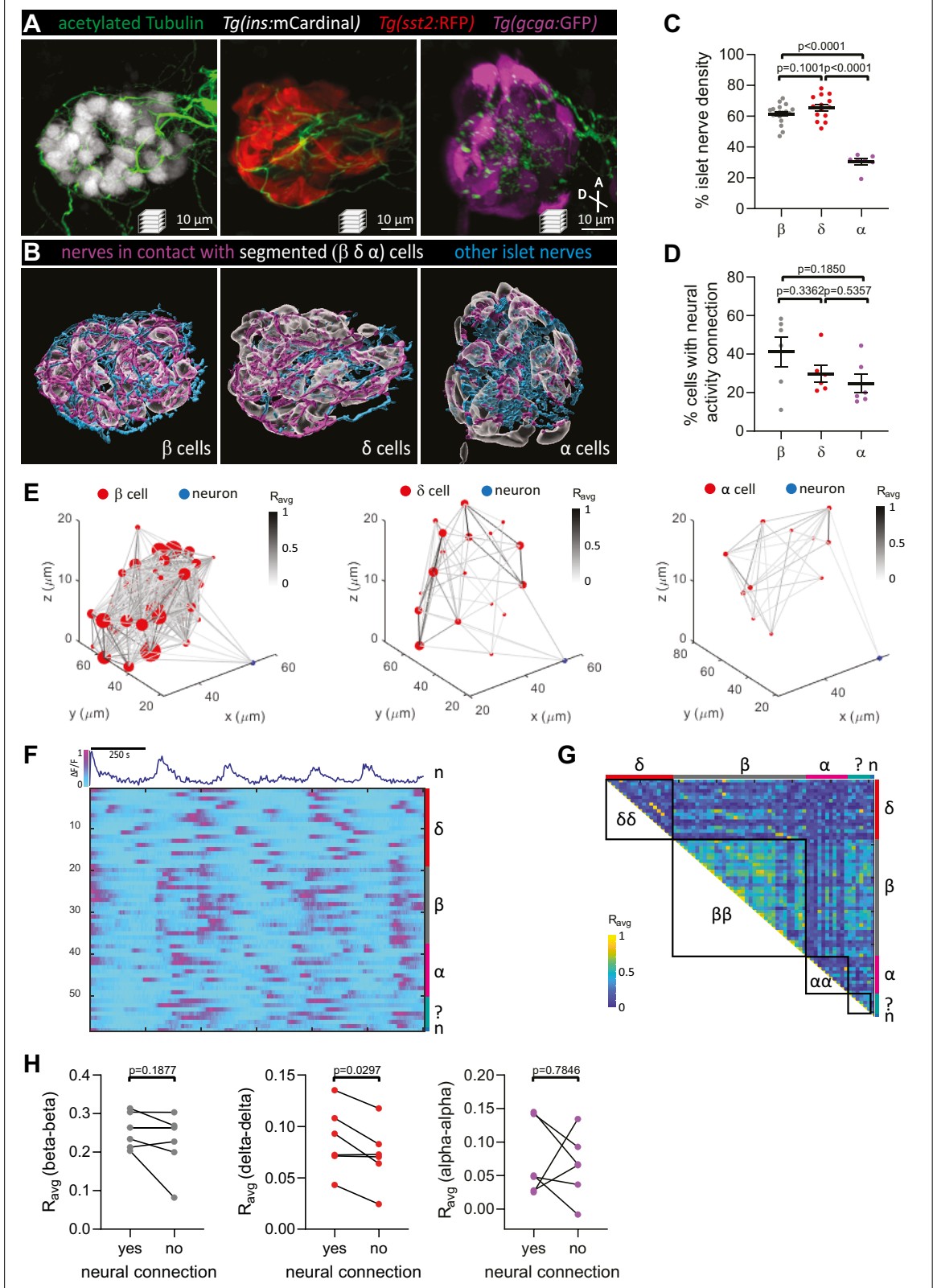

**Figure 2.** Pancreatic nerves display differential interactions with islet cell types.
 (**A**) 100 hpf *Tg(ins:mCardinal)*, *Tg(sst2:RFP)*, and *Tg(gcga:GFP)* zebrafish immuno-stained for acetylated Tubulin (nerves). (**B**) Segmentation and classification of islet nerves that are in contact with beta, delta, or alpha cell surfaces (magenta). All remaining islet nerves are colored in cyan. (**C**) Percentage of islet nerve density in contact with the specified pancreatic islet cell type; a majority of the nerves are in contact with beta and delta

*Figure 2 continued on next page*

*Figure 2 continued*

cells; mean ± standard error of the mean (SEM), $n$ = 7–18 animals, p values from one-way analysis of variance (ANOVA) with Holm–Sidak's multiple comparisons test; see *Figure 2—source data 1*. (**D**) Following exposure to elevated glucose and calcium imaging of pancreatic islet cells and peri-islet neurons at 105–110 hpf, correlation analysis was used to identify beta, delta, and alpha cells with neural activity connection. The percentage of cells of a given type with neural activity connection shows no significant difference between the different cell types; mean ± SEM, $n$ = 6 animals, p values from one-way ANOVA with Holm–Sidak's multiple comparisons test; see *Figure 2—source data 2*. (**E**) Correlation maps of a peri-islet neuron and beta, delta, and alpha cells. Individual cells are plotted with their coordinates. The strength of the correlation ($R_{avg}$) between cell pairs is drawn with a grayscale line and the number of connections for each cell is represented by the circle size. (**F**) Normalized calcium traces of pancreatic islet cells (including delta, beta, alpha, and unidentified cells) and a neuron (n). Individual islet cells were assigned to a cell type and given a cell id. (**G**) Correlation matrix of cell activity. Individual cells were assigned to a cell type (including delta, beta, alpha, and unidentified cells, and neurons) and average correlation coefficients for given cell pairs (matrix row-column intersects) were calculated. Areas displaying homotypic interactions are highlighted. (**H**) Average correlation coefficients from individual animals between homotypic cell pairs were divided into two groups (cells with or without neural connection); $n$ = 6 animals, p values from paired *t*-tests.

The online version of this article includes the following source data for figure 2:

**Source data 1.** *Figure 2C*.

**Source data 2.** *Figure 2D*.

chronic neural inhibition (*Figure 3I*), suggesting that neural signaling is an important regulator of alpha–beta connectivity.

Given the potential role of pancreatic innervation on islet cell maturation, we next investigated the effects of acutely blocking neural activity using two different approaches. By lineage tracing, we found that the neural crest-derived peri-islet neurons were also labeled by the *Et(1121A:GAL4FF)* enhancer trap (*Figure 4A, B*), thereby allowing us to investigate the effects of photo-ablating a subset of peri-islet neurons on islet cell activity (*Figure 4C*). The photo-ablation of peri-islet neurons reduced islet nerve density (*Figure 4—figure supplement 1A, B*). While the oscillatory pattern of calcium dynamics was maintained in beta cells upon this ablation (*Figure 4D*), the coupling between beta cells was significantly decreased (*Figure 4E–G*). Notably, we did not observe further impairment in beta cell coupling over increasing distance (*Figure 4G*), suggesting that upon ablation of peri-islet neurons, the signal that initiates beta cell coupling was blunted while beta cells maintained their propensity for coupling across the islet. Whether this beta cell coupling is due to gap junctional (*Benninger et al., 2008*) and/or soluble factors warrants further studies. Delta–beta coupling was also reduced (*Figure 4—figure supplement 1C*). Unlike in the chronic neural inhibition scenario, in the fraction time analysis, delta–beta and alpha–delta coupling was not affected upon ablating peri-islet neurons (*Figure 4H–J*). However, we observed a decrease in alpha–beta coupling and in the alpha-active/beta-active state (*Figure 4G–H*). The observed coupling defects are unlikely due to changes in individual calcium spike characteristics, as no significant differences were observed in calcium oscillation frequency, peak height, or peak duration (*Figure 4—figure supplement 2*). Overall, the ablation of peri-islet neurons significantly disrupted beta–beta and alpha–beta connectivity, while conclusions regarding other heterotypic interactions will require further investigation into the various neural subsets that were targeted. It is likely that our targeting of peri-islet neurons affected at least those that guide the activity coupling between alpha and beta cells.

Next, we took an optogenetic approach by generating a transgenic line that allows one to acutely photo-inhibit the release of neurotransmitters with a single pulse of blue light. This method has previously been used in *Drosophila* (*Makhijani et al., 2017*) and *C. elegans* (*Lin et al., 2013*; *Qi et al., 2012*) for targeted photo-ablation and photo-inhibition. Pan-neural expression of a singlet oxygen generator, miniSOG2, tethered to synaptic granules resulted in a blue light inducible loss of swimming activity in 110 hpf larvae (*Figure 5—figure supplement 1A–C*). Following this confirmation of the effectiveness of the tool, we studied neural control of islet cell activity upon acute photo-inhibition (*Figure 5—figure supplement 1D*). Surprisingly, photo-inhibition decreased glucose levels compared with transgene-negative zebrafish exposed to the same light condition (*Figure 5A*). Similar to peri-islet neural ablation, pan-neural photo-inhibition decreased beta cell connectivity (*Figure 5B–E*). Changes in delta–beta and alpha–beta heterotypic interactions were also observed upon acute neural inhibition (*Figure 5E–H*). A significant decrease in delta-silent/beta-active state reflects what we observed upon chronic neural inhibition (*Figure 5H*). Like with the photo-ablation of peri-islet neurons, no changes in nearest alpha–delta interactions were observed (*Figure 5F–H*). Notably, following acute

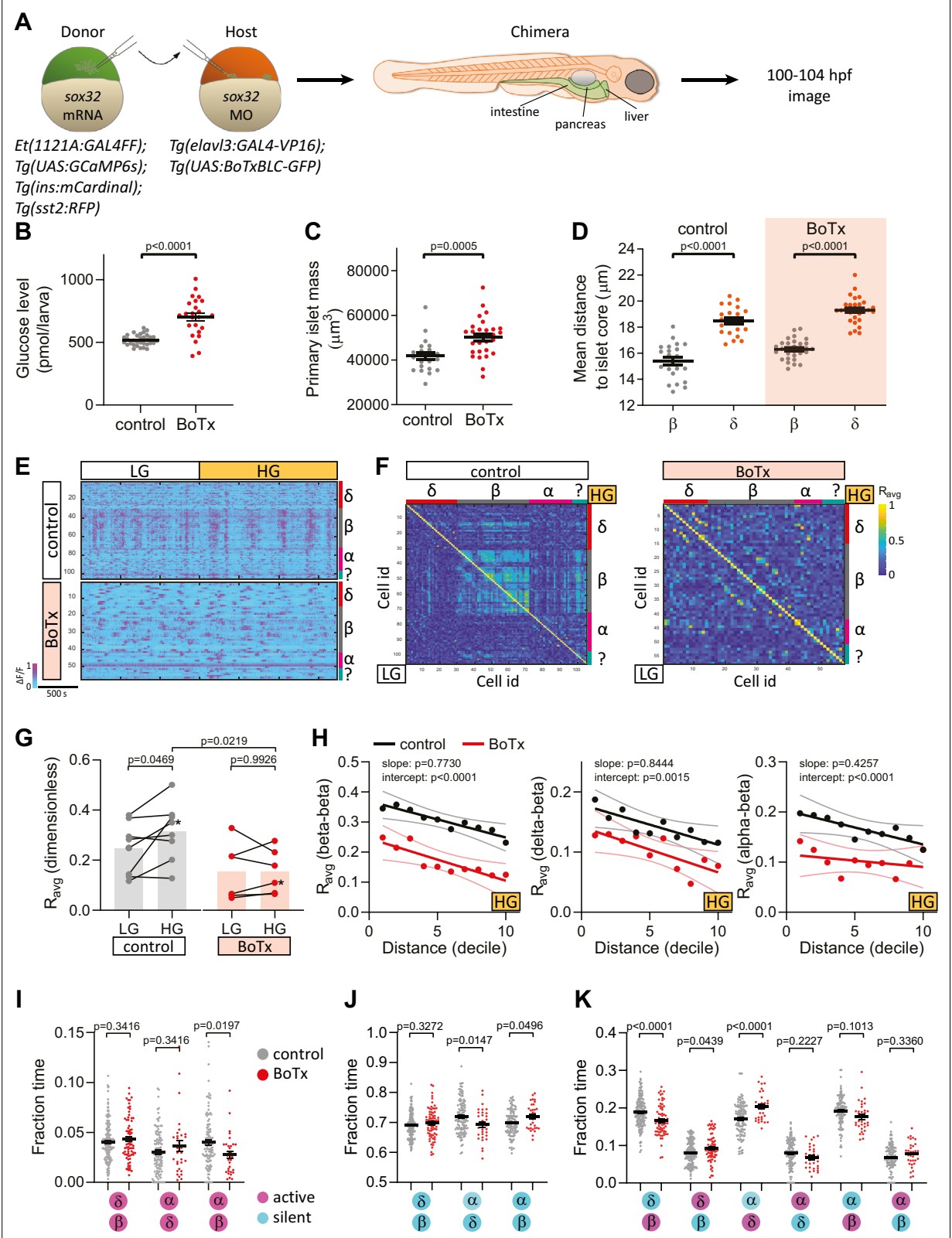

**Figure 3.** Chronic inhibition of synaptic transmission disrupts islet cell activity. (**A**) Schematic of transplants to generate chimeras with endodermal organs derived entirely from donor embryos. (**B**) Whole larva-free glucose-level measurements at 100 hpf; mean ± standard error of the mean (SEM), n = 24–32 batches of 5 larvae per replicate, p value from *t*-test; see *Figure 3—source data 1*. (**C**) Quantification of primary islet mass; n = 21–29, p value from *t*-test; see *Figure 3—source data 2*. (**D**) Mean distance of pancreatic islet cells to islet core; mean ± SEM, n = 21–29 animals, p values

*Figure 3 continued on next page*

*Figure 3 continued*

from paired *t*-tests; see *Figure 3—source data 3*. (**E**) Normalized calcium traces of pancreatic islet cells (including delta, beta, alpha, and unidentified cells). Individual islet cells were assigned to a cell type and given a cell id; LG, basal condition; HG, glucose treated condition. (**F**) Correlation matrices of islet cell activity. Individual islet cells were assigned to a cell type and given a cell id, and average correlation coefficients for given cell pairs (matrix row-column intersects) were calculated for LG (basal condition; bottom left triangle) and HG (glucose treated condition; top right triangle). (**G**) Average beta cell correlation coefficients in individual larvae; *n* = 5–8 animals, p values from two-way analysis of variance (ANOVA) with Holm–Sidak's multiple comparisons test; *, corresponding calcium traces and correlation matrices shown in panels E and F. (**H**) Average homotypic (beta–beta) and heterotypic (delta–beta and alpha–beta) cell correlation coefficients with cell distance distribution from 1 (close) to 10 (far), mean and linear regression (solid lines) with 95% confidence intervals; *n* = 5–8 animals, p values of slope and intercept from simple linear regression. (**I–K**) Fraction time analysis of heterotypic delta–beta, alpha–delta, and alpha–beta cell pairs for times when both are active (**I**), both are silent (**J**), and one is active and one is silent (**K**); mean ± SEM, *n* = 32–160 cell pairs, p values from two-way ANOVA with Holm–Sidak's multiple comparisons test; magenta circle, active state; cyan circle, silent state.

The online version of this article includes the following source data and figure supplement(s) for figure 3:

**Source data 1.** *Figure 3B*.

**Source data 2.** *Figure 3C*.

**Source data 3.** *Figure 3D*.

**Figure supplement 1.** Glucose treatment elevates whole larva glucose levels.

**Figure supplement 2.** Islet cell activity following chronic inhibition of synaptic transmission.

**Figure supplement 3.** Chronic inhibition of synaptic transmission does not affect calcium peak frequency, height, or duration in delta cells.

---

neural inhibition, alpha–beta coupling (*Figure 5E*) and alpha-active/beta-active states (*Figure 5F*) were significantly decreased. While we cannot exclude a role for soluble factors from other peripheral organs, these changes in alpha–beta interactions were consistently observed upon both acute pan-neural and peri-islet inhibition, possibly reflecting a role for neurons in the maintenance of alpha–beta coupling.

Dissecting the complex interplay of local autocrine, paracrine, and gap junctional communication between different endocrine cells, in addition to vascular and nerve interactions, is often hindered by our inability to simultaneously study them in an intact organ within its innate environment. Imaging calcium dynamics with genetically encoded biosensors or calcium sensitive fluorescent indicators in individual islet cell types has been conducted in vitro with dispersed cells (*Yang et al., 2013*; *Albrecht et al., 2015*, whole islets *Benninger et al., 2008*; *Johnston et al., 2016*), and perfused pancreas slices (*Stožer et al., 2013*; *Panzer et al., 2020*), and in vivo with islets transplanted into the anterior chamber of the eye (*Rodriguez-Diaz et al., 2012*; *Salem et al., 2019*), as well as intravital imaging of the mouse pancreas (*Adams et al., 2021*). We report a noninvasive in vivo imaging strategy to study all the different pancreatic endocrine cell types within the same animal. Our three approaches to inhibit neural control, ranging in temporal and spatial specificity, provided useful insights into the role of neurons in regulating pancreatic islet function (*Figure 6*). Whether activity coupling between beta cells is in part mediated by gap junctions warrants further studies, but our data suggest that neural regulation is critical for the establishment and maintenance of beta cell connectivity as we consistently found decreased beta cell coupling upon chronic and acute neural inhibition. Given that our targeted neural ablation approach also led to this decline in beta cell coupling, it is likely that autonomic neural control is required for beta cell connectivity independently of possible indirect effects resulting from neural regulation of other organs. Whether this neural regulation is directly on beta cells or a bystander effect resulting from the regulation of other cell types, including delta and endothelial cells, requires further studies. It has been proposed that glucose sensing neurons regulate early postnatal beta cell proliferation and maintenance of beta cell function (*Tarussio et al., 2014*), and our data support the peri-islet localization of such neurons.

We have focused our studies on the pancreatic beta, alpha, and delta cells; however, it is important to note that there are other endocrine cell types (gamma and epsilon cells) that remain undefined, but do display glucose induced activity coupling with beta cells (*Figure 3F*). Correlation analysis allowed us to study heterotypic coupling and fraction time analysis further allowed us to study activity patterns of heterotypic cell pairs that are near each other, upon loss of neural signaling. We found changes in delta–beta activity coupling, supporting a role for gap junctions in mediating electrical coupling between delta and beta cells (*Briant et al., 2018*). Both chronic and acute neural inhibition also

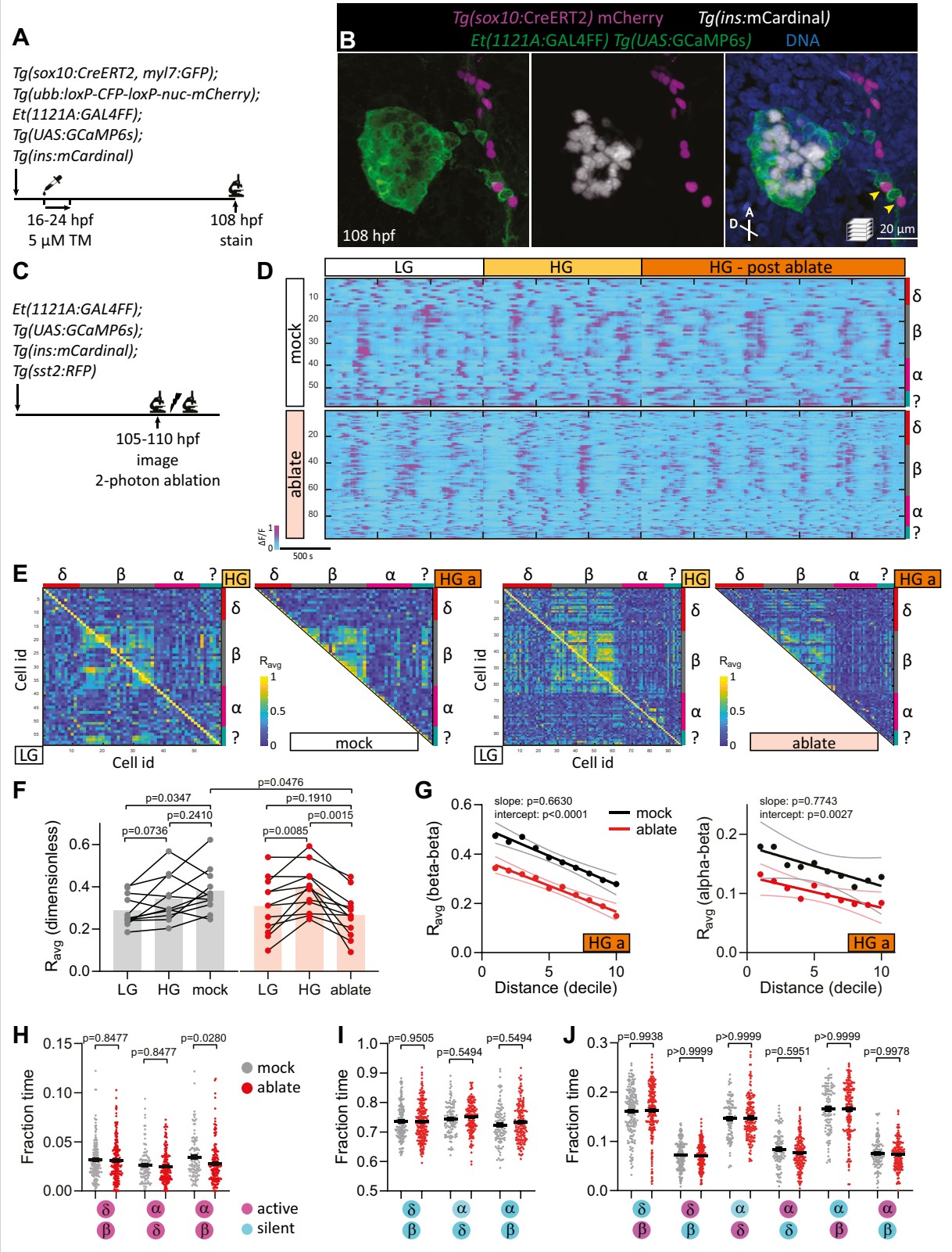

**Figure 4.** Targeted ablation studies reveal the crucial role of peri-islet neurons for islet cell activity. (**A**) Schematic of lineage tracing of neural crest cells in *Tg(sox10:CreERT2, myl7:GFP); Tg(ubb:loxP-CFP-loxP-nuc-mCherry); Tg(ins:mCardinal); Et(1121A:GAL4FF); Tg(UAS:GCaMP6s)* zebrafish following 5 µM tamoxifen (TM) treatment from 16 to 24 hpf and staining at 108 hpf. (**B**) Wholemount immunostaining at 108 hpf for mCherry expression (neural crest-derived cells) and counterstaining with DAPI (DNA). Yellow arrowheads point to neural crest-derived cells positive for GCaMP6s expression. (**C**)

*Figure 4 continued on next page*

*Figure 4 continued*

Schematic of two-photon ablation experiment. (**D**) Normalized calcium traces of pancreatic islet cells (including delta, beta, alpha, and unidentified cells). Individual islet cells were assigned to a cell type and given a cell id; LG, basal condition; HG, glucose treated condition; HG a, ablation or mock ablation condition. (**E**) Correlation matrices of islet cell activity; LG, basal condition; HG, glucose treated condition; HG a, ablation or mock ablation condition. (**F**) Average beta cell correlation coefficients in individual larvae; $n$ = 11–12 animals, p values from two-way analysis of variance (ANOVA) with Holm–Sidak's multiple comparisons test. (**G**) Average homotypic (beta–beta) and heterotypic (alpha–beta) cell correlation coefficients with cell distance distribution from 1 (close) to 10 (far), mean and linear regression (solid lines) with 95% confidence intervals; $n$ = 11–12 animals, p values of slope and intercept from simple linear regression. Fraction time analysis of heterotypic delta–beta, alpha–delta, and alpha–beta cell pairs for times when both are active (**H**), both are silent (**I**), and one is active and one is silent (**J**); mean ± standard error of the mean (SEM), $n$ = 178–215 cell pairs, p values from two-way ANOVA with Holm–Sidak's multiple comparisons test; magenta circle, active state; cyan circle, silent state.

The online version of this article includes the following figure supplement(s) for figure 4:

**Figure supplement 1.** Targeted ablation of peri-islet neurons affects islet nerve density and islet cell activity.

**Figure supplement 2.** Targeted ablation of peri-islet neurons does not affect calcium peak frequency, height, or duration in islet cells.

blunted the functional connectivity between alpha and beta cells suggesting that neurons may have an important role in the paracrine potentiation of beta cell activity by glucagon released from alpha cells (*Svendsen et al., 2018*). We propose that alpha–beta interactions are specifically regulated by pancreatic innervation since peri-islet neural ablation led to a similar decrease in alpha–beta coupling and the alpha-active/beta-active state. However, given that pan-neural inhibition is required to induce changes in nearest delta–beta interactions, it is possible that central nervous control of other organs could in part be driving these changes. Importantly, our data further support a role for neurons in modulating delta cell activity, since delta cells that display neural activity connection exhibit a significant increase in their coupling to other delta cells. Future studies will determine whether the observed changes in delta cell activity directly reflect alterations in somatostatin release.

Selective targeting of subsets of neurons will advance our understanding of the pancreatic islet-neural interplay in health and disease, including diabetes pathophysiology. Through our in vivo studies of homotypic and heterotypic activity coupling, we have illustrated how studying functional connectivity can be achieved for the endocrine pancreas and discovered a critical role for neurons in mediating these connections. Given the cellular heterogeneity of organ composition, simultaneously evaluating the function of the different cell types that make up an organ provides insights that can be missed by simply investigating one cell type at a time. Interrogating the neural regulation of other organs, such as the liver, intestine, and kidney, can be achieved through the extended application of these tools. In combination with the ability to monitor organ development, function, and regeneration in vivo, this approach will allow one to address complex questions pertaining to the autonomic nervous system and its role in organ maintenance and dysfunction.

## Materials and methods

### Zebrafish transgenic lines and husbandry

All zebrafish husbandry was performed under standard conditions in accordance with institutional (MPG) and national ethical and animal welfare guidelines (Proposal numbers: B2/1041, B2/Anz. 1007, B2/1218). All procedures conform to the guidelines from Directive 2010/63/EU of the European Parliament on the protection of animals used for scientific purposes. Adult zebrafish were fed a combination of fry food (Special Diet Services) and brine shrimp five times daily and maintained under a light cycle of 14 hr light:10 hr dark at 28.5°C. Zebrafish embryos and larvae were grown in egg water at 28.5°C. Transgenic and mutant lines used were on the *mitfa*$^{w2/w2}$ background and as described in *Table 1*.

### Generation of new transgenic lines

The *Tg(elavl3:sypb-miniSOG2-P2A-mScarlet)* line was generated by Tol2 transgenesis of a 8.7 kb *elavl3* promoter (Addgene: 59531, AgeI restriction enzyme digested) driving expression of *sypb* (Addgene: 74316), *miniSOG2* (Addgene: 87410), *P2A-mScarlet* (gift from A. Beisaw) cloned with Cold Fusion (System Biosciences). The *Tg(ins:mCardinal)* line was generated by Tol2 transgenesis of a 1.1 kb *ins* promoter (in-house plasmid, MluI restriction enzyme digested) driving expression of *mCardinal* (Addgene: 51311) cloned with Cold Fusion.

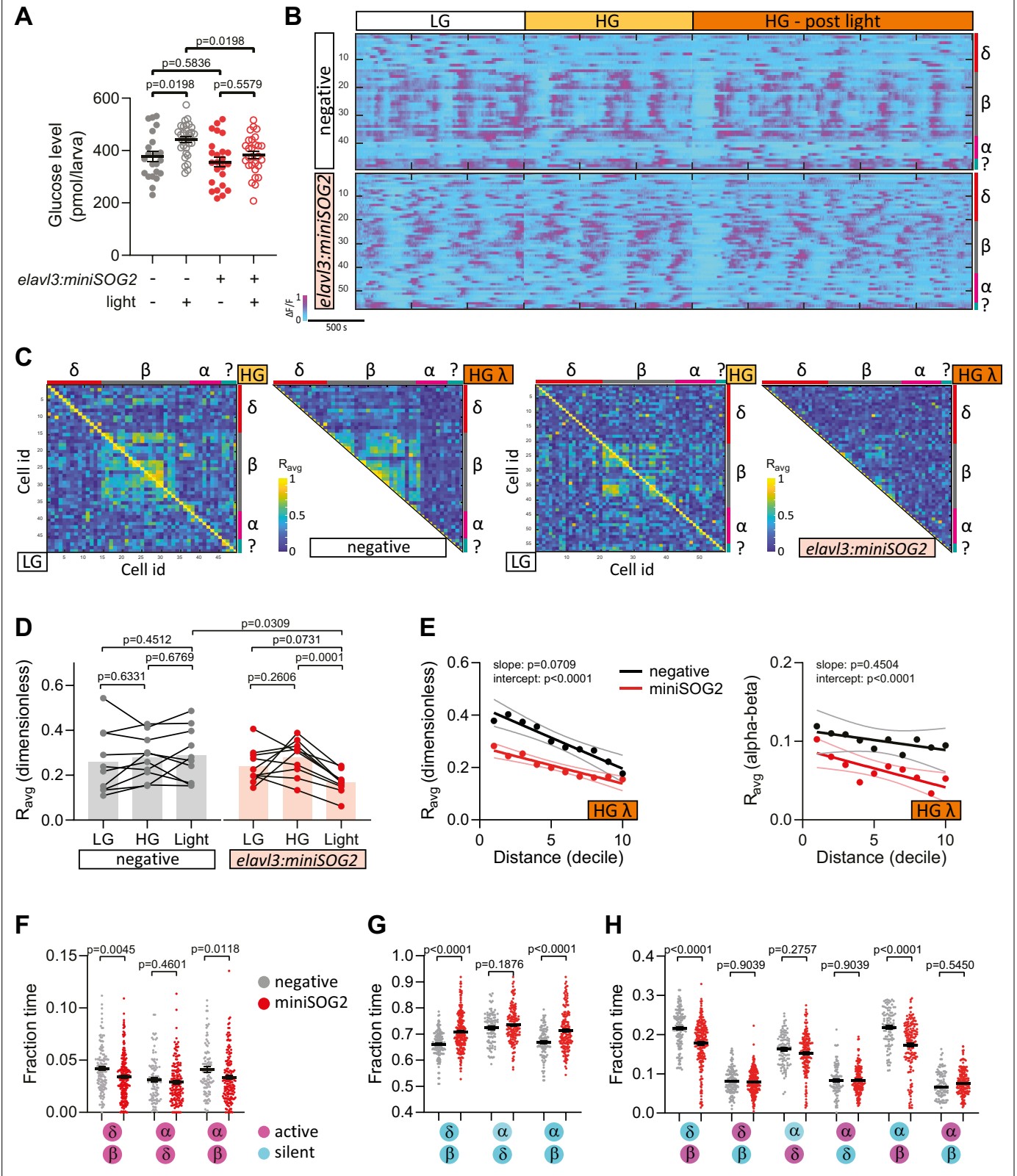

**Figure 5.** Acute optogenetic inhibition of neurotransmitter release disrupts islet cell activity. (**A**) Whole larva-free glucose-level measurements of *Tg(elavl3:sypb-miniSOG2-P2A-mScarlet)* zebrafish at 110 hpf; mean ± standard error of the mean (SEM), *n* = 22–31 batches of 5 larvae per replicate, p values from two-way analysis of variance (ANOVA) with Holm–Sidak's multiple comparisons test; see *Figure 5—source data 1*. (**B**) Normalized calcium traces of pancreatic islet cells (including delta, beta, alpha, and unidentified cells). Individual islet cells were assigned to a cell type and given a cell id;

*Figure 5 continued on next page*

*Figure 5 continued*

LG, basal condition; HG, glucose treated condition. (**C**) Correlation matrices of islet cell activity; LG, basal condition; HG, glucose treated condition; HG $\lambda$, post-blue light treatment. (**D**) Average beta cell correlation coefficients in individual larvae; *n* = 10 animals, p values from two-way ANOVA with Holm–Sidak's multiple comparisons test. (**E**) Average homotypic (beta–beta) and heterotypic (alpha–beta) cell correlation coefficients with cell distance distribution from 1 (close) to 10 (far), mean and linear regression (solid lines) with 95% confidence intervals; *n* = 10 animals, p values of slope and intercept from simple linear regression. Fraction time analysis of heterotypic delta–beta, alpha–delta, and alpha–beta cell pairs for times when both are active (**F**), both are silent (**G**), and one is active and one is silent (**H**); mean ± SEM, *n* = 141–220 cell pairs, p values from two-way ANOVA with Holm–Sidak's multiple comparisons test; magenta circle, active state; cyan circle, silent state.

The online version of this article includes the following source data and figure supplement(s) for figure 5:

**Source data 1.** *Figure 5A*.

**Figure supplement 1.** Swimming activity is reduced following blue light exposure in *Tg(elavl3:sypb-miniSOG2-P2A-mScarlet)* larvae.

**Figure supplement 2.** Islet cell activity following acute optogenetic inhibition of neurotransmitter release.

## Generation of chimeric zebrafish

*Et(1121A:GAL4FF); Tg(UAS:GCaMP6s); Tg(sst2:RFP); Tg(ins:mCardinal)* donor embryos were injected with 200 pg *sox32* mRNA, generated with mMESSAGE mMACHINE SP6 transcription kit (Thermo), at the one-cell stage to enhance endoderm formation. *Tg(elavl3:GAL4-VP16); Tg(UAS:BoTxBLC-GFP)* host embryos were injected with 0.3 ng *sox32* morpholino (Gene Tools LLC) at the one-cell stage to inhibit endoderm formation. At the 1000-cell stage, single cells were removed from donor embryos with a glass capillary needle, and ~40 cells were transplanted into host embryos in the region fated to give rise to endoderm-derived tissues. Only larvae that displayed complete endoderm transplant, without obvious defects, and carried the relevant transgenes were used for downstream imaging experiments. Control chimeras were negative for BoTxBLC-GFP expression.

## In vivo confocal microscopy

Live zebrafish between 100 and 110 hpf were anesthetized with 0.015% Tricaine and mounted in 0.8% low melting agarose in egg water containing 0.005% Tricaine for confocal imaging. A Zeiss LSM880 upright laser scanning confocal microscope equipped with a Plan-Apochromat ×20/NA1.0 dipping lens was used for imaging. Time-lapse calcium imaging was conducted in 25°C conditions and z-stacks were taken at 5-s intervals for 1–2 hr. Larvae were exposed to 75 mM glucose containing egg water at the indicated time points during the time-lapse imaging to increase whole larvae glucose levels within a physiological range (*Figure 3—figure supplement 1*). For the photo-inhibition experiments,

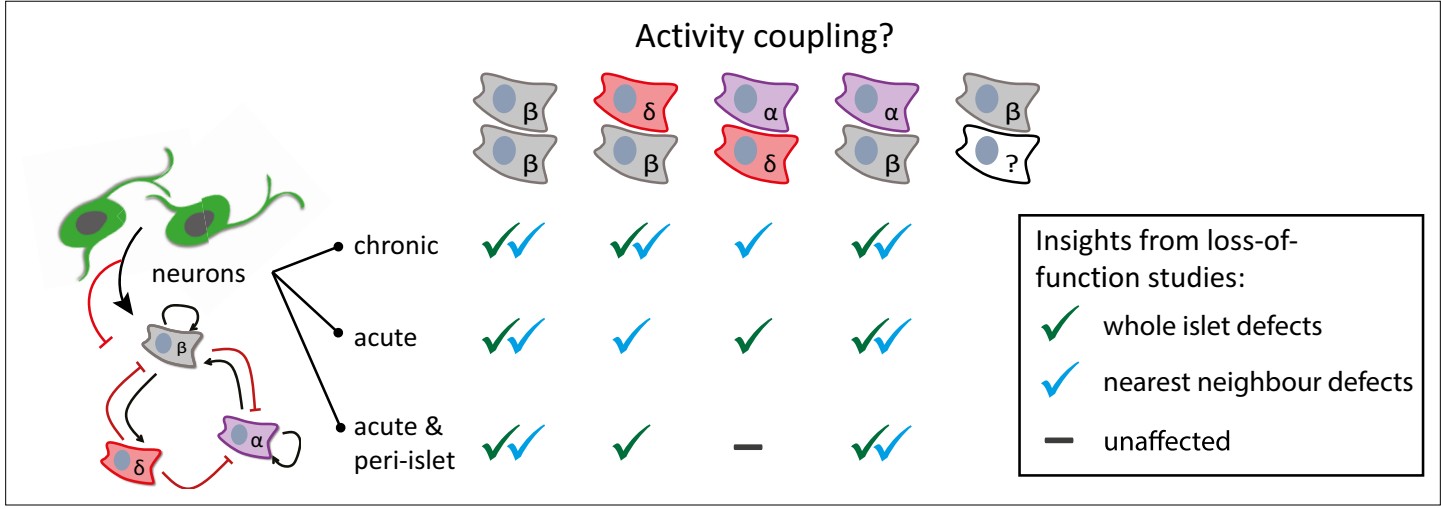

**Figure 6.** Graphical abstract summarizing the research findings. Neuromodulation, with varying temporal and spatial specificity, provides useful insights into the role of neurons in regulating homotypic and heterotypic activity coupling between pancreatic islet cells. Green check marks indicate when significant defects were observed in correlation-based analysis across the whole islet. Blue check marks indicate when significant defects were observed in fraction time analysis between nearest neighbors.

**Table 1.** List of zebrafish transgenic lines.

| Name | Specificity/purpose | Reference |
|---|---|---|
| *Tissue specific promoter lines* | | |
| *Et(1121A:GAL4FF)*[nkgsaizgffm1121A] | Pancreatic islet cells, neurons | Current |
| *Tg(ins:mCardinal)*[bns162] | Pancreatic beta cells | Current |
| *Tg(elavl3:sypb-miniSOG2-P2A-mScarlet)*[bns529] | Inhibit postmitotic neurons | Current |
| *Tg(elavl3:GAL4-VP16)*[zf357] | Postmitotic neurons | *Stevenson et al., 2012* |
| *Tg(sst2:RFP)*[gz19] | Pancreatic delta cells | *Li et al., 2009* |
| *Tg(gcga:GFP)*[ia1] | Pancreatic alpha cells | *Zecchin et al., 2007* |
| *Tg(sox10:CreERT2, myl7:GFP)*[t007] | Neural crest cells, heart marker | *Mongera et al., 2013* |
| *Tg(ubb:loxP-CFP-loxP-nuc-mCherry)*[jh63] | Ubiquitous | *Wang et al., 2015* |
| *Tg(kdrl:EGFP)*[s843] | Endothelial cells | *Jin et al., 2005* |
| *UAS lines* | | |
| *Tg(UAS:GCaMP6s)*[UAShspzGCaMP6s] | Visualize intracellular calcium | *Muto et al., 2017* |
| *Tg(UAS:BoTxBLC-GFP)*[icm21] | Inhibit neurotransmitter release | *Yang et al., 2018*; *Sternberg et al., 2016* |

the GCaMP6s signal was visualized with a Chameleon Vision II Ti:Sapphire Laser (Coherent) laser at 920 nm. For all other experiments, the GCaMP6s signal was visualized with an Argon laser at 488 nm.

## Two-photon laser ablation

A Chameleon Vision II Ti:Sapphire Laser (Coherent) mounted on a Zeiss LSM880 microscope was used for two-photon single-cell laser ablation of peri-islet neurons labeled by the *Et(1121A:GAL4FF)* enhancer trap. The tunable laser was set at 800 nm to scan an ablation area of 4 $\mu m^2$ at a scan speed of 1 with 10 iterations. Following ablation, embryos were rested for 5 min before continuation of calcium imaging. Controls were mock ablation of cells within 20 µm from the peri-islet neurons. Animals from the same clutch were randomly assigned to ablation and control groups.

## Photo-inhibition of nerve activity

All experiments with *Tg(elavl3:sypb-miniSOG2-P2A-mScarlet)* animals were done on F1 to F3 larvae displaying strong and uniform mScarlet signal. Controls were transgene-negative siblings. To assess swimming behavior, 110 hpf larvae were individually isolated in 9-mm diameter circular wells filled with 200 µl egg water. A Nikon SMZ25 stereomicroscope with a P2-SHR PlanApo ×1/NA0.15 objective was used for time-lapse imaging of larvae at 1-s intervals; the same animals were imaged prelight and postexposure to a 5-min pulse of blue light (3.8 mW LED, 466/40 filter, Lumencor Sola Light Engine). To assess changes in glucose levels, pools of 40 larvae swimming in 9 cm diameter Petri dishes filled with 35 ml egg water were exposed to 0.3 mW blue LED light for 30 min prior to sample collection; control animals were not exposed to blue light and were randomly selected siblings from the same clutch. Notably, acute blue light exposure alone mildly reduced swimming behavior and increased glucose levels (*Figure 5—figure supplement 1C*, *Figure 5A*). Induction of a stress response likely led to this increase in glucose levels in wild-type larvae, as chronic 14 hr exposure to blue LED light for 3 consecutive days kills zebrafish larvae (*Ustundag et al., 2019*). To assess calcium changes, larvae were exposed to a 3-min pulse of blue light (2.1 mW 470 nm LED, Colibri) and the GCaMP6s signal was measured pre- and post-blue light exposure. While blinding was not possible, we randomized the order in which animals on a given day were imaged.

## Wholemount immunostaining

Zebrafish were euthanized with Tricaine overdose prior to overnight fixation in 4% paraformaldehyde dissolved in phosphate-buffered saline (PBS) containing 120 µM $CaCl_2$ and 4% sucrose, pH 7.4. The skin was manually removed with forceps, without disturbing the internal organs and the zebrafish were permeabilized with 1% Triton X-100, 1% dimethyl sulfoxide (DMSO) containing PBS

for 3 hr at room temperature. Following blocking with 5% donkey serum (Jackson Immunoresearch) in blocking buffer (Dako), samples were incubated in primary antibodies overnight at 4°C, washed 4× with 0.025% Triton X-100 containing PBS, incubated in secondary antibodies overnight at 4°C, washed 4×, incubated in an increasing glycerol gradient of 25%, 50%, and 75%, and mounted in VectorShield mounting medium. The following antibodies and dilutions were used: mouse anti-glucagon (1:200, Sigma G2654), chicken anti-GFP (1:200, Aves GFP-1020), mouse anti-acetylated Tubulin (1:200, Sigma T7451), and guinea pig anti-insulin (1:300, DAKO A0564). Secondary antibodies used in this study include donkey anti-guinea pig AlexaFluor647 (1:200, Jackson 706-605-148), donkey anti-mouse AlexaFluor488 (1:300, Jackson 715-545-150) and AlexaFluor647 (1:200, Jackson 715-605-150), and donkey anti-chicken AlexaFluor488 (1:300, Jackson 703-545-155). Nuclei were stained with 25 µg/ml diamidino-2-phenylindole (DAPI, Sigma). Images were taken on a Zeiss LSM880 or LSM800 laser scanning confocal microscope equipped with a ×25/NA0.8 objective.

## Data analysis

Image data were analyzed using Imaris (Bitplane) and Fiji (ImageJ). Correlation and fraction time analyses were performed using Matlab (code available upon request). Statistical analysis was performed using Prism 8 (GraphPad) and the type of test used is indicated in the figure legends. For comparison between two groups, a two-tailed Student's $t$-test was used to determine the p values. For comparison between more than two groups, an ordinary one-way analysis of variance ANOVA with Holm–Sidak's multiple comparisons test or a two-way ANOVA with Holm–Sidak's multiple comparisons test was used to determine the p values. The number of animals or cells analyzed, and the p values are reported in the figure and figure legends. All experiments were repeated on different days using at least three different clutches of animals.

## Acknowledgements

These studies were supported by funds from the Max Planck Society to DYRS. YHCY was supported by CIHR Postdoctoral Fellowships, an HFSP Long-Term Fellowship (LT000159/2015), an EMBO Long-Term Fellowship (ALTF 773-2014), NIG-JOINT funding, and UKRI Expanding Excellence in England. LJBB was supported by a Sir Henry Wellcome Postdoctoral Fellowship. KK was supported by an NBRP grant from AMED. We thank Arica Beisaw for the plasmid with *P2A-mScarlet* and James Johnson for critical reading of the manuscript.

## Additional information

### Competing interests

Koichi Kawakami: Reviewing editor, *eLife*. Didier YR Stainier: Senior editor, eLife. The other authors declare that no competing interests exist.

### Funding

| Funder | Grant reference number | Author |
| --- | --- | --- |
| Max Planck Society | | Didier YR Stainier |
| Canadian Institutes of Health Research | | Yu Hsuan Carol Yang |
| Human Frontier Science Program | LT000159/2015 | Yu Hsuan Carol Yang |
| EMBO | ALTF 773-2014 | Yu Hsuan Carol Yang |
| NIG-JOINT | | Yu Hsuan Carol Yang |
| AMED | | Koichi Kawakami |
| Wellcome Trust | | Linford JB Briant |

| Funder | Grant reference number | Author |
|---|---|---|
| UKRI Expanding Excellence in England | | Yu Hsuan Carol Yang |

The funders had no role in study design, data collection, and interpretation, or the decision to submit the work for publication.

## Author contributions

Yu Hsuan Carol Yang, Conceptualization, Data curation, Formal analysis, Funding acquisition, Investigation, Methodology, Project administration, Resources, Software, Supervision, Validation, Visualization, Writing – original draft, Writing – review and editing; Linford JB Briant, Formal analysis, Software, Writing – review and editing; Christopher A Raab, Formal analysis, Investigation; Sri Teja Mullapudi, Methodology, Writing – review and editing; Hans-Martin Maischein, Methodology; Koichi Kawakami, Resources, Writing – review and editing; Didier YR Stainier, Funding acquisition, Resources, Writing – review and editing

## Author ORCIDs

Yu Hsuan Carol Yang ⓘ http://orcid.org/0000-0001-6663-0302
Linford JB Briant ⓘ http://orcid.org/0000-0003-3619-3177
Sri Teja Mullapudi ⓘ http://orcid.org/0000-0002-3916-8148
Koichi Kawakami ⓘ http://orcid.org/0000-0001-9993-1435
Didier YR Stainier ⓘ http://orcid.org/0000-0002-0382-0026

## Ethics

All zebrafish husbandry was performed under standard conditions in accordance with institutional (MPG) and national ethical and animal welfare guidelines (Proposal numbers: B2/1041, B2/Anz. 1007, B2/1218). All procedures conform to the guidelines from Directive 2010/63/EU of the European Parliament on the protection of animals used for scientific purposes.

## Decision letter and Author response

Decision letter https://doi.org/10.7554/eLife.64526.sa1
Author response https://doi.org/10.7554/eLife.64526.sa2

# Additional files

## Supplementary files

• Transparent reporting form

## Data availability

All data generated or analysed during this study are included in the manuscript, figures, and figure legends. Source data files have been provided for Figures 1, 2, 3, and 5.

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
