## [Editor Report]

The role of islet innervation on endocrine cell activity is currently not well defined. Previously, Yang et al. described embryonic islet innervation dynamics in zebrafish and the role of neural activity in proper glucose homeostasis and δ-cell formation. In their new manuscript, the authors introduced a novel transgenic model for whole islet in vivo calcium imaging. By applying this tool in a series of innovative and technically challenging approaches, the authors provide novel insights into how neuronal inputs influence pancreatic endocrine cell connectivity and function. Overall, this is an important study that adds to our understanding for the role of neuronal interactions with the islet.

---

## [Decision Letter]

**Decision letter after peer review:**

Thank you for submitting your article "Innervation modulates the functional connectivity between pancreatic endocrine cells" for consideration by *eLife*. Your article has been reviewed by 3 peer reviewers, and the evaluation has been overseen by a Reviewing Editor and Marianne Bronner as the Senior Editor. The following individuals involved in review of your submission have agreed to reveal their identity: Sarah M Knox (Reviewer #3); Dirk Meyer (Reviewer #4).

The reviewers have discussed the reviews with one another and the Reviewing Editor has drafted this decision to help you prepare a revised submission.

The role of islet innervation on endocrine cell activity is currently not well defined. Previously, Yang et al. provided a detailed description of embryonic islet innervation dynamics in zebrafish and they established that innervation and/or neural activity is required for proper glucose homeostasis and δ-cell formation. In their new manuscript, they introduced a novel transgenic model for whole islet in vivo calcium imaging. By applying this tool in a series of innovative and technically challenging approaches, the authors employ 3 distinct methods to modulate neuronal activity using a range of reporter lines, mutant lines and intravital imaging to examine these islet cell interactions. In addition they demonstrate that larval β-cells express the gap-junction protein Cx43 and that β-cell coupling is disturbed in cx43 mutants. On the whole these methods are robustly applied, in addition to the analytical methods used to interpret 'connectivity' from the ca^2+^ dynamics. Overall, this is an important study that adds to our currently weak understanding for the role of neuronal interactions with the islet.

Summary:

Overall, the reviewers are supportive of publication and study provides important novel insights into our understanding of the underlying mechanisms mediating coordinated pancreatic endocrine cell responses. However, they indicate the need for a number of important clarifications (particularly related to interpreting the connectivity analysis) and places where data presentation is not always clear. In addition, all 3 reviewers indicated that throughout the manuscript the authors' made overly speculative statements and there is some concern that the study does not provide evidence of direct influence of neuronal activity. Finally, some of the experiments require additional and/or alternative statistical analyses. While most of these revisions will not require additional experimentation, a small number of experiments are suggested that would help support the authors' conclusions.

Essential revisions:

1. An important caveat in this study is that connectivity is used to refer to cells showing similar ca^2+^ elevations, whether calculated via correlation coefficients or fraction of time two neighboring cells are 'active'/'inactive'. This does not per se indicate if the two cells truly are directly interacting, or whether they are instead each interacting with some other factor. For instance, an increased correlation coefficient between β cells upon innervation could be due to enhanced gap junction coupling or paracrine communication between β cells or it could be due to neuronal activity driving the activity of the β cells. Some literature (e.g. Zhang et al., Biophys.J. 2008) would suggest the latter, not the former, and thus conflicts with the authors conclusions.

2. Related to this above point, in figures 2-4 the authors use one type of analysis for assessing homotypic interactions between β cells, but a separate analysis for assessing heterotypic interactions. Why do the authors not use a correlation based analysis between cell neighbors for heterotypic interactions? Would they observe changes if such correlation analysis is employed? (I see why they cannot use it across all cells).

Also, if the 'connectivity' between two cell types changes (e.g. β-δ), then the apparent connectivity with the other cell type (α) would very likely change irrespective of whether any actual connection is present. For instance, it seems to me that changes in β cell correlation would change the apparent β-δ and β-α connectivity, irrespective of the actual link between these cell types. Δ-α 'connectivity' doesn't change under any of the experimental conditions which would support this alternative conclusion. How do the authors exclude this alternative conclusion?

Finally, are α, β and δ cell activity patterns themselves altered under each neuronal modulation and Cx43 deletion? E.g. the duration and amplitude of ca^2+^ transients. If so could these explain differences in apparent 'connectivity'?

3. There are many overly speculative statements that need either toning down or including data that better supports the conclusion: e.g.

– Abstract: "δ cells receive innervation which coordinates its (the islet) output." Such an interpretation would rely on modulating neuronal-δ cell interactions specifically.

– Lines 123-124: hyper-activation of δ cells is a strong statement. Instead, it seems that δ cells are activated more than their neighboring β and α cells. But the change in δ cell activity is not especially strong. Is δ cell activity itself (% time elevated or spike height) different?

– Line 124: similarly, the authors did not measure somatostatin release so cannot say this is directly the result of δ cell activity.

– Line 192-194: the authors should combine their Cx43 with neural modulation to test if pancreatic innervation is indeed mediating the remaining intra-islet coordination.

4. In performing neuronal modulation, pan-neural chronic botox treatment (Figure 2) significantly disrupted both homotypic and heterotypic interactions; acute photo-ablation of peri-islet neurons (Figure 3) only disrupted homotypic interactions (although note my comment 1); and pan-neural acute inhibition of neurotransmitter release (Figure 4) significantly disrupted both homotypic and heterotypic interactions. How can the authors exclude the pan-neural disruptions affecting islet function independent of pancreas innervation, e.g. innervation affecting another organ and changing some circulating factor? Especially given both pan neural methods generate different findings than the peri-islet disruption.

5. Line 140-141: you cannot say there is a trend towards a decrease in α-active β active state based on a p of 0.06 within a set of data that contains 12 related comparisons. This leads me to ask if the authors have corrected for multiple comparisons in their statistical analysis? (for 2J-L, for 3I-K, for4G-I). Throughout, the authors should state in the figure captions what statistical analysis was performed for which comparison indicated.

6. Figure 2 G: the change in Ravg (correlation) is marginally significant. The results refer to these changes as 'significantly disrupted', 'severe disruption', drastically altered' which needs toning down, with 'marginal' being a more apt term. Furthermore, in F the 'representative' correlation matrices are not congruent with the data in G. At the very least in G the data points corresponding to the example in F should be indicated, and better would be to include more representative matrices.

7. In figures 2-4 I do not follow the Ravg vs distance measurement. This is interpreted to be indicating β cell wave propagation. I do not see how the decline in relative correlation with distance would refer to wave propagation? This is really a decorrelation which is entirely expected for a system that is not perfectly coupled. Other phenomena than wave propagation could explain such decorrelation. I suggest the authors either analyze the time-index of Ca across the islet to specifically show wave propagation, or refer to this data appropriately.

8. While the islet cell type specific reporters coupled with calcium imaging represents a novel and exciting new technology for understanding zebrafish endocrine functionality, the authors utilize it along with three methods to address the same question; namely, do nerves regulate islet cell functional connectivity. Of the three, one method (photo-ablation; Figure 3), shows contradictory/weak results when compared to the other two and is thus not warranted as a main figure and could be moved to supplement.

9. In figure 3 (photo ablation of peri-islet nerves) the authors show no evidence that photo ablation does indeed lead to non-functional/disrupted nerve architecture. To address this, the authors could fix and stain for nerves directly after photo ablation to see if neuronal axons are still present.

10. While the findings of cx43 KO leading to impaired β cell functionality are novel, they have no direct tie to the previous 4 figures of the paper. The relationship to the nerves is supported by solid data, indeed, there is very, very little data to indicate nerve input plays a role here.

11. The authors state that δ cell activity is increased in the chronic and acute models of neuronal inhibition (BoTx and optogenetics), suggestion that nerves play an inhibitory role in δ cell function. This, however, cannot be proven directly by the current experiments performed by the authors. Furthermore, the current experiments cannot definitively say that the nerves play a direct role in regulating neuronal activity vs. bystander effect of modulating other surrounding cell types (i.e. vasoconstriction of vascularization within the islet).

12. The authors cannot attribute which branch of the peripheral nervous system (parasympathetic vs. sympathetic) are leading to functional disruptions upon inhibition. While botox is predominantly thought of as an inhibitor of cholinergic neurons, there is evidence that administration affects pre-ganglionic ganglia that activate sympathetic neurons. Furthermore, optogenetic inhibition of pancreatic nerves will potentially inhibit both sympathetic and parasympathetic nerves. Thus, determining which peripheral nerve subtype is important for functional signaling in pancreatic islet cell would be novel and beneficial to the field.

13. Does the BoTx affect the health of the zebrafish, which might affect the function of the pancreatic islet cells? This needs to be addressed.

14. While the authors also shed new light on the previously unclear role of gap-junctions in the zebrafish islet and β-cell coupling, it remains unclear how these results connect to the innervation phenotypes. Rather, it might have been more interesting to use Et(1121A:LAL4FF) driven GCaMP expression in peri-islet neurons to directly correlate neuronal with endocrine activity – in particular with respect to the suggested role of neural input in modulating δ cell function.

15. 109, Figure 2G: The data show no significant difference between LG – HG, suggesting that there is no glucose-induced response, same for Figure 3F and 4D (such a difference is only seen for the control in 5E ?). In addition: paired sample t-test rather than student's t-test should be used for comparing LG and HG (3F: repeated measures ANOVA?)

16. 109-111 "[…] indicating drastically altered synchronicity and wave propagation (Figure 2H-I)." – data in H show no difference in slopes between both groups, purpose/meaning of AOC is not explained and relevance appears unclear as there is no significant change.

17. 123-125: "[…] resulted in the hyperactivation of δ cells, […]" – Figure 2J-L: These conclusions need more detailed statistics, in particular as it is not clear which statistical tests were used – information about sample size, standard deviation and or confidence intervals, degrees of freedom, t-value are missing. (same for corresponding data in Figure 3-5).

18. 183-184: "From normalized single cell calcium traces, we observed blunted

glucose-induced coupling of β cell activity glucose-induced coupling" – see above: it seems this is the only data set supporting a glucose response in control animals?

19. 520: "[…]; p-values from ANOVA." P-values are from post-hoc test? Holm Sidak's multiple comparisons should be applied instead.

---

## [Author Response]

Summary:Overall, the reviewers are supportive of publication and study provides important novel insights into our understanding of the underlying mechanisms mediating coordinated pancreatic endocrine cell responses. However, they indicate the need for a number of important clarifications (particularly related to interpreting the connectivity analysis) and places where data presentation is not always clear. In addition, all 3 reviewers indicated that throughout the manuscript the authors' made overly speculative statements and there is some concern that the study does not provide evidence of direct influence of neuronal activity. Finally, some of the experiments require additional and/or alternative statistical analyses. While most of these revisions will not require additional experimentation, a small number of experiments are suggested that would help support the authors' conclusions.

We thank the reviewers for their supportive and constructive comments. We have conducted additional experiments and analyses and have made the recommended changes in toning down our language. The revised Figure 2 provides further support for direct neural regulation of a subset of pancreatic islet cells. We have addressed all of the reviewers’ comments as detailed below.

Essential revisions:1. An important caveat in this study is that connectivity is used to refer to cells showing similar ca^2+^ elevations, whether calculated via correlation coefficients or fraction of time two neighboring cells are 'active'/'inactive'. This does not per se indicate if the two cells truly are directly interacting, or whether they are instead each interacting with some other factor. For instance, an increased correlation coefficient between β cells upon innervation could be due to enhanced gap junction coupling or paracrine communication between β cells or it could be due to neuronal activity driving the activity of the β cells. Some literature (e.g. Zhang et al., Biophys.J. 2008) would suggest the latter, not the former, and thus conflicts with the authors conclusions.

We thank the reviewers for this comment and have rephrased the sentences that implicated direct interactions. We have now also cited the suggested paper.

2. Related to this above point, in figures 2-4 the authors use one type of analysis for assessing homotypic interactions between β cells, but a separate analysis for assessing heterotypic interactions. Why do the authors not use a correlation based analysis between cell neighbors for heterotypic interactions? Would they observe changes if such correlation analysis is employed? (I see why they cannot use it across all cells).

We have now also included a correlation-based analysis for heterotypic interactions and presented the results alongside the fraction time analysis.

Also, if the 'connectivity' between two cell types changes (e.g. β-δ), then the apparent connectivity with the other cell type (α) would very likely change irrespective of whether any actual connection is present. For instance, it seems to me that changes in β cell correlation would change the apparent β-δ and β-α connectivity, irrespective of the actual link between these cell types. Δ-α 'connectivity' doesn't change under any of the experimental conditions which would support this alternative conclusion. How do the authors exclude this alternative conclusion?

While we cannot exclude the alternative conclusion completely, we think that the differential effects observed across the different conditions that all display β cell correlation changes support the presence of connectivity between heterotypic cell pairs. For instance, there is some evidence of δ-α connectivity changes in both chronic and acute pan-neural inhibition condition, but not in the acute, peri-islet ablation setting. In these ablation experiments, only α-β changes were concurrently observed for both the correlation and fraction time analysis.

Finally, are α, β and δ cell activity patterns themselves altered under each neuronal modulation and Cx43 deletion? E.g. the duration and amplitude of ca^2+^ transients. If so could these explain differences in apparent 'connectivity'?

We have now included the analysis of duration and amplitude of the calcium transients for the ablation condition. There doesn’t appear to be any significant differences in frequency, duration, or amplitude of calcium transients for β, δ, or α cells across the different conditions and upon ablation.

3. There are many overly speculative statements that need either toning down or including data that better supports the conclusion: e.g.– Abstract: "δ cells receive innervation which coordinates its (the islet) output." Such an interpretation would rely on modulating neuronal-δ cell interactions specifically.– Lines 123-124: hyper-activation of δ cells is a strong statement. Instead, it seems that δ cells are activated more than their neighboring β and α cells. But the change in δ cell activity is not especially strong. Is δ cell activity itself (% time elevated or spike height) different?– Line 124: similarly, the authors did not measure somatostatin release so cannot say this is directly the result of δ cell activity.– Line 192-194: the authors should combine their Cx43 with neural modulation to test if pancreatic innervation is indeed mediating the remaining intra-islet coordination.

We have toned down the language.

As noted by the reviewers, further analysis of δ cell activity did not show any significant difference in calcium peak frequency, height, or duration (Figure 3 – Figure Suppl. 3). We have therefore removed the claims of δ cell hyper-activation.

We thank the reviewers for this suggestion. While the proposed *cx43* experiments would certainly be very interesting to carry out, it was not feasible given the COVID19 related and lab startup delays. We have therefore decided to remove the *cx43* data.

4. In performing neuronal modulation, pan-neural chronic botox treatment (Figure 2) significantly disrupted both homotypic and heterotypic interactions; acute photo-ablation of peri-islet neurons (Figure 3) only disrupted homotypic interactions (although note my comment 1); and pan-neural acute inhibition of neurotransmitter release (Figure 4) significantly disrupted both homotypic and heterotypic interactions. How can the authors exclude the pan-neural disruptions affecting islet function independent of pancreas innervation, e.g. innervation affecting another organ and changing some circulating factor? Especially given both pan neural methods generate different findings than the peri-islet disruption.

We cannot exclude the possibility that the pan-neural disruption affects the pancreas indirectly, and have now added a sentence on this point.

5. Line 140-141: you cannot say there is a trend towards a decrease in α-active β active state based on a p of 0.06 within a set of data that contains 12 related comparisons. This leads me to ask if the authors have corrected for multiple comparisons in their statistical analysis? (for 2J-L, for 3I-K, for4G-I). Throughout, the authors should state in the figure captions what statistical analysis was performed for which comparison indicated.

Our study with the pan-neural ablation was under powered, and we have now increased the number of biological replicates. The change in α-β heterotypic interaction is statistically significant, both with the correlation analysis and fraction time analysis, suggesting that innervation does play a role in α-β interactions.

Information about the statistical analysis performed is now included in all the figure legends.

6. Figure 2 G: the change in Ravg (correlation) is marginally significant. The results refer to these changes as 'significantly disrupted', 'severe disruption', drastically altered' which needs toning down, with 'marginal' being a more apt term. Furthermore, in F the 'representative' correlation matrices are not congruent with the data in G. At the very least in G the data points corresponding to the example in F should be indicated, and better would be to include more representative matrices.

The language has been toned down.

The correlation matrices in panel F match the calcium traces in panel E. For panel G, we have now indicated the data points corresponding to the examples in panels E & F.

7. In figures 2-4 I do not follow the Ravg vs distance measurement. This is interpreted to be indicating β cell wave propagation. I do not see how the decline in relative correlation with distance would refer to wave propagation? This is really a decorrelation which is entirely expected for a system that is not perfectly coupled. Other phenomena than wave propagation could explain such decorrelation. I suggest the authors either analyze the time-index of Ca across the islet to specifically show wave propagation, or refer to this data appropriately.

We have rephrased our interpretation as recommended and removed all reference to wave propagation.

8. While the islet cell type specific reporters coupled with calcium imaging represents a novel and exciting new technology for understanding zebrafish endocrine functionality, the authors utilize it along with three methods to address the same question; namely, do nerves regulate islet cell functional connectivity. Of the three, one method (photo-ablation; Figure 3), shows contradictory/weak results when compared to the other two and is thus not warranted as a main figure and could be moved to supplement.

The photo-ablation experiments add a level of specificity to targeting pancreatic innervation. Although, we don’t yet know the exact identity of these neurons, more animals have since been analyzed to increase the power of our analysis.

9. In figure 3 (photo ablation of peri-islet nerves) the authors show no evidence that photo ablation does indeed lead to non-functional/disrupted nerve architecture. To address this, the authors could fix and stain for nerves directly after photo ablation to see if neuronal axons are still present.

To support the claim of disrupted islet nerve architecture upon photo-ablation of peri-islet neurons (now Figure 4), we have included quantification of islet nerve density following peri-islet neuron ablation in Figure 4 —figure supplement 1A-B.

10. While the findings of cx43 KO leading to impaired β cell functionality are novel, they have no direct tie to the previous 4 figures of the paper. The relationship to the nerves is supported by solid data, indeed, there is very, very little data to indicate nerve input plays a role here.

We thank the reviewer for this comment and have decided to remove the *cx43* data from this manuscript.

11. The authors state that δ cell activity is increased in the chronic and acute models of neuronal inhibition (BoTx and optogenetics), suggestion that nerves play an inhibitory role in δ cell function. This, however, cannot be proven directly by the current experiments performed by the authors. Furthermore, the current experiments cannot definitively say that the nerves play a direct role in regulating neuronal activity vs. bystander effect of modulating other surrounding cell types (i.e. vasoconstriction of vascularization within the islet).

We have toned down the language. Additionally, and as suggested by the reviewers, we have now analyzed the activity of peri-islet neurons in parallel with islet cells (new Figure 2) and the data do suggest that a subset of δ cells may be directly modulated by nerve activity. We cannot exclude bystander effects and have included a statement to that effect in the discussion.

12. The authors cannot attribute which branch of the peripheral nervous system (parasympathetic vs. sympathetic) are leading to functional disruptions upon inhibition. While botox is predominantly thought of as an inhibitor of cholinergic neurons, there is evidence that administration affects pre-ganglionic ganglia that activate sympathetic neurons. Furthermore, optogenetic inhibition of pancreatic nerves will potentially inhibit both sympathetic and parasympathetic nerves. Thus, determining which peripheral nerve subtype is important for functional signaling in pancreatic islet cell would be novel and beneficial to the field.

We have previously demonstrated that at the stages when these experiments were carried out, innervation is only from the vagus nerve and have since found that sympathetic innervation does not enter the islet until after 120 hpf. We have now included this point in the results and plan to analyze effects of sympathetic nerves with appropriate transgenic lines.

13. Does the BoTx affect the health of the zebrafish, which might affect the function of the pancreatic islet cells? This needs to be addressed.

Pan-neural expression of BoTx does have other effects that may influence pancreatic islet function. As we have previously described (PMID: 29916364), these fish may also have defects in the development of δ cells. This point has now been mentioned in the results.

14. While the authors also shed new light on the previously unclear role of gap-junctions in the zebrafish islet and β-cell coupling, it remains unclear how these results connect to the innervation phenotypes. Rather, it might have been more interesting to use Et(1121A:LAL4FF) driven GCaMP expression in peri-islet neurons to directly correlate neuronal with endocrine activity – in particular with respect to the suggested role of neural input in modulating δ cell function.

We thank the reviewers for this comment. We have removed the *cx43* data and added examples of correlation of neural activity with endocrine activity (new Figure 2). Notably, we found that δ cells displaying significant activity coupling to a peri-islet neuron had significantly higher activity coupling to other δ cells within the same islet in comparison to those that were not coupled to the neuron.

15. 109, Figure 2G: The data show no significant difference between LG – HG, suggesting that there is no glucose-induced response, same for Figure 3F and 4D (such a difference is only seen for the control in 5E ?). In addition: paired sample t-test rather than student's t-test should be used for comparing LG and HG (3F: repeated measures ANOVA?)

The statistics have now been updated and described in the figure legends. There is a significant difference between LG – HG in the control of the BoTx experiment (Figure 3G). However as noted by the reviewers, this glucose induced response is not observed in Figure 4F and 5D. The photo-inhibition/photo-ablation experiments had to be carried out 5 hours later than the BoTx experiment, due to the inability to photo-ablate the neurons at earlier stages. The ~20% rise in glucose level in the earlier stages (Figure 3- Figure Suppl. 1) had a significant effect on β cell correlation. The increased basal glucose level in the later stages (PMID: 23201900) likely led to less prominent induction of glucose, and thus a lesser effect on β cell correlation.

16. 109-111 "[…] indicating drastically altered synchronicity and wave propagation (Figure 2H-I)." – data in H show no difference in slopes between both groups, purpose/meaning of AOC is not explained and relevance appears unclear as there is no significant change.

This statement has now been removed since, as the reviewers pointed out, our analyses do not allow one to infer any changes in wave propagation.

17. 123-125: "[…] resulted in the hyperactivation of δ cells, […]" – Figure 2J-L: These conclusions need more detailed statistics, in particular as it is not clear which statistical tests were used – information about sample size, standard deviation and or confidence intervals, degrees of freedom, t-value are missing. (same for corresponding data in Figure 3-5).

We have removed this statement as further analysis of δ cell activity did not reveal any significant difference in calcium peak frequency, height, or duration (Figure 3 – Figure Suppl. 3). Figure legends have been updated.

18. 183-184: "From normalized single cell calcium traces, we observed bluntedglucose-induced coupling of β cell activity glucose-induced coupling" – see above: it seems this is the only data set supporting a glucose response in control animals?

The section on *cx43* has been removed.

19. 520: "[…]; p-values from ANOVA." P-values are from post-hoc test? Holm Sidak's multiple comparisons should be applied instead.

Appropriate post-hoc tests were performed and described in the figure legends and methods.